POINT OF VIEW

# Bioengineering horizon scan 2020

**Abstract** Horizon scanning is intended to identify the opportunities and threats associated with technological, regulatory and social change. In 2017 some of the present authors conducted a horizon scan for bioengineering (Wintle et al., 2017). Here we report the results of a new horizon scan that is based on inputs from a larger and more international group of 38 participants. The final list of 20 issues includes topics spanning from the political (the regulation of genomic data, increased philanthropic funding and malicious uses of neurochemicals) to the environmental (crops for changing climates and agricultural gene drives). The early identification of such issues is relevant to researchers, policy-makers and the wider public.

LUKE KEMP*, LAURA ADAM, CHRISTIAN R BOEHM, RAINER BREITLING, ROCCO CASAGRANDE, MALCOLM DANDO, APPOLINAIRE DJIKENG, NICHOLAS G EVANS, RICHARD HAMMOND, KELLY HILLS, LAUREN A HOLT, TODD KUIKEN, ALEMKA MARKOTIĆ, PIERS MILLETT, JOHNATHAN A NAPIER, CASSIDY NELSON, SEÁN S ÓHÉIGEARTAIGH, ANNE OSBOURN, MEGAN J PALMER, NICOLA J PATRON, EDWARD PERELLO, WIBOOL PIYAWATTANAMETHA, VANESSA RESTREPO-SCHILD, CLARISSA RIOS-ROJAS, CATHERINE RHODES, ANNA ROESSING, DEBORAH SCOTT, PHILIP SHAPIRA, CHRISTOPHER SIMUNTALA, ROBERT DJ SMITH, LALITHA S SUNDARAM, ERIKO TAKANO, GWYN UTTMARK, BONNIE C WINTLE, NADIA B ZAHRA AND WILLIAM J SUTHERLAND*

*For correspondence: ltk27@cam. ac.uk (LK); w.sutherland@zoo.cam. ac.uk (WJS)

## Introduction

Bioengineering is expected to have profound impacts on society in the near future as applications increase across multiple areas, while costs and barriers to access fall. The speed of this change and the breadth of the applications make the task of forecasting the impacts of bioengineering both urgent and difficult (*Guston, 2014*). In 2017 we published the results of a 'horizon scan' that looked at emerging issues in bioengineering (*Wintle et al., 2017*). Here we report the results of an updated horizon scan based on a wider range of inputs (38 participants from six continents and 13 countries, compared with 27 participants from the UK and US in the 2017 exercise) and a broader definition of bioengineering.

We followed the same structured 'investigate, discuss, estimate and aggregate' (IDEA) protocol for identifying and prioritising issues (*Hanea et al., 2017*), with some minor adjustments (see Methods). We tasked our experts with identifying 'novel, plausible and high-impact' issues in biological engineering, and they produced a long list of 83 issues. Participants then scored the issues anonymously (with a score out of 1,000, reflecting likelihood, impact and novelty), arriving at a short list of 41 issues to be discussed at a workshop. This was coupled with a 'yes/no' question to determine whether the issues were novel, based on whether the experts had heard of the issue previously. After deliberation, participants re-scored these issues. The issues identified in the latest horizon scan differ substantially from those identified in 2017. This change likely stems from an increase in the diversity of the participants, improvements in the methods used, a broader definition of bioengineering, and changes in the research landscape since 2017.

Since it was undertaken, there have been developments in a number of the issues identified in the 2017 bioengineering horizon scan. Human germline genome editing came to prominence in late 2019 when researcher He Jiankui announced the birth of two girls with CRISPR/Cas9-edited genomes (*Cyranoski, 2019*). Military funding of bioengineering projects also remained substantial: for example, projects funded by DARPA included programs to explore the use bioelectronics for tissue repair and regeneration (BETR) and to develop mosquito-repellent skin (ReVector). There have also been breakthroughs in the use of enhanced photosynthesis for agricultural productivity: a 2018 study reported that metabolic engineering strategies increased photosynthetic efficiency by 17%, which resulted in an increase of about 20% in biomass in field conditions (*South et al., 2019*). This technology is now being deployed in several crops. The use of 'platform technologies to address emerging disease pandemics', another topic identified in 2017, has taken on particular significance as a result of the COVID-19 pandemic. Many of the vaccine candidates for COVID-19 currently undergoing clinical and preclinical evaluation have been developed from platforms for non-coronavirus candidates such as influenza, SARs and Ebola (*WHO, 2020*).

Horizon scanning aims to build societal preparedness by systematically identifying upcoming opportunities and threats from technological, regulatory and social change (*Sutherland and Woodroof, 2009*). Horizon scanning with the Delphi technique has a long history. It has been used to identify emerging critical issues in areas as diverse as conservation biology (*Sutherland et al., 2006*; *Sutherland et al., 2017*), invasive species in the UK (*Ricciardi et al., 2017*), poverty reduction (*Pretty et al., 2010*) and biosecurity (*Boddie et al., 2015*). Periodic horizon scanning is also undertaken in some areas: in global conservation, for example, these scans have identified issues such as micro-plastics, gene editing for invasive species, and cultivated meat approximately six years before they captured public attention (*Sutherland et al., 2017*). Horizon-scanning activities related to the Antarctic and Southern Ocean (*Kennicutt et al., 2014*) have also directed funding and policy (*Kennicutt et al., 2019*), and helped to provide

the basis for research roadmaps (*Kennicutt et al., 2015*).

In this article we provide a high-level summary of the top 20 issues identified in the bioengineering horizon scan 2020 (while acknowledging that the number of topics covered means that there will be some sacrifice of depth for breadth). We take a broader view of bioengineering than we did in 2017, defining it as the application of ideas, principles and techniques to the engineering of biological systems. This means that we now cover more aspects of bioengineering, as well as issues that contribute to or result from bioengineering advances (such as funding). To avoid giving a false sense of forecasting precision or overemphasising minor differences in scoring, the issues are not ranked, and are instead grouped into issues that are expected to be most relevant within five years, within 5–10 years, and on timescales of longer than 10 years (*Table 1*). Our intent is to spur further research into these issues and further discussion of their implications by researchers, policy-makers and the wider public.

## The issues most relevant within five years

### Access to biotechnology through outsourcing

Traditionally, the biotechnology sector has had high barriers to entry, with organizations needing to build extensive physical and knowledge-based assets. New 'cloud labs' and services labs are circumventing this model by using technologies such as robotics, automation and the internet to offer widely-accessible standardised services with limited need for physical material transfer (*Jessop-Fabre and Sonnenschein, 2019*). This facilitates both broader access and faster development of new products through the sharing of capital and knowledge across projects (*Lentzos and Invernizzi, 2019*). It is also helping to empower non-traditional researchers by lowering the threshold for participating in cutting-edge research.

This distributed approach poses a biosecurity gap as research activities are separated from intent: the cloud lab may not seek additional details on an experiment's context, including why it is being performed. There is also a lack of appropriate biosecurity guidelines and governance models to handle this (*Palmer et al.,*

**Table 1.** Overview of the bioengineering horizon scan 2020.

Summary of the 20 issues identified through the scan; issues are grouped according to likely timeline for realisation.

| <5 Years | 5–10 Years | >10 Years |
|---|---|---|
| Access to biotechnology through outsourcing | Agricultural gene drives | Bio-based production of materials |
| Crops for changing climates | Neuronal probes expanding new sensory capabilities | Live plant dispensers of chemical signals |
| Function-based design in protein engineering | Distributed pharmaceutical development and manufacturing | Malicious use of advanced neurochemistry |
| Philanthropy shapes bioscience research agendas | Genetically engineered phage therapy | Enhancing carbon sequestration |
| State and international regulation of DNA database use | Human genomics converging with computing technologies | Porcine bioengineered replacement organs |
| | Microbiome engineering in agriculture | The governance of cognitive enhancement |
| | Phytoremediation of contaminated soils | |
| | Production of edible vaccines in plants | |
| | The rise of personalised medicine such as cell therapies | |

2015; *Dunlap and Pauwels, 2019*). As outsourcing through cloud labs becomes increasingly prevalent in the next five years, these challenges may require the development of new guidelines and business and incentive models for responsible innovation and biosecurity.

### Crops for changing climates

Climate change is predicted to result in more frequent droughts and intensive precipitation events. This will increase soil salinity, elevate average temperatures, and shift the range, abundance and genotypic diversity of pollinators, pests and pathogens. All of these factors are expected to impact crop yields. In response, efforts are intensifying to adapt food production using agro-ecological strategies (*Altieri et al., 2015*), as well the provision of well-adapted crop varieties by genetic engineering and new breeding technologies (*Dhankher and Foyer, 2018*): Drought-tolerant genetically modified (GM) plant varieties have reached the market and more are in development (*Nuccio et al., 2018*); the capabilities of plant immune receptors have been broadened by protein engineering (*De la Concepcion et al., 2019*); and the identification of conserved submergence-activated genes has revealed novel genetic targets for enhancing flood tolerance (*Reynoso et al.,*

2019). Technical progress is still required for success in the field. However, deployment may be hindered by a comparative lack of funding for plant science, as well as lengthy and expensive regulatory regimes in most jurisdictions. New models for public-private co-operation will be needed to advance the translation of basic research through to the field, including business models that are not based on simple economic returns. The effects of novel traits on biodiversity and ecosystems will require further scrutiny before being deployed in a warmer world.

### Function-based design in protein engineering

Despite a growing understanding of the relationship between protein structure and function, efficient design of new proteins with a desired action has remained a laborious process. For example, chimeric antigen receptor (CAR) thymus lymphocyte (T cell) therapies which combine functional protein moieties to activate T cells against malignant tumours have only recently been approved for human use after decades of iteration (*Feins et al., 2019*). The convergence of ongoing developments, including substantial improvements in predicting protein structure from amino acid sequences using machine learning (*AlQuraishi, 2019*; *Yang et al., 2019a*),

could overcome previous technical and computational challenges. This indicates a potential revolution in function-based protein design, leading to various useful industrial compounds (such as the development of catalysts for any desired organic reaction) and medical applications (such as the ability to selectively destroy, suppress or stimulate any malfunctioning tissue, which is the key to treating many refractory diseases). However, as this field grows so will the risk of deliberate misuse. Protein engineering could be used to produce agents that have a higher lethality or specificity than existing agents (including new agents based on novel mechanisms of action). Protein engineering might also simplify the production of toxins currently derived from natural sources.

### Philanthropy shapes bioscience research agendas

Over the past decade, philanthropic funding (including venture philanthropy) of research and innovation has been increasing (*Coutts, 2019*; *Depecker et al., 2018*). This has largely been driven by the increasing concentration of wealth, and erosion of public health and scientific research initiatives within key countries. These investments can provide particular research groups or areas with substantial funding over prolonged periods of time, and they can also support areas of research that are not usually funded by governments. Philanthropic investments can also promote innovation, such as allowing for more exotic approaches not usually funded by governments. However, these investments might also influence the development of biotechnologies in a way that has less of a public mandate than government-funded research. Philanthropic investments also operate without the traditional mechanisms for accountability, transparency or oversight often required by federal or state law (*Reich, 2018*). Some areas of medical research are already considerably underfunded compared to health needs (*Rafols and Yegros, 2018*), and philanthropic investments may exacerbate this discrepancy in the near-term future. Significant investment into a small range of actors could also undermine diversity, particularly at the international level (*Lentzos, 2019*). A possible response would be partnerships between public and private investors, though such partnerships might not fully address concerns about accountability, transparency or oversight.

### State and international regulation of deoxyribonucleic acid (DNA) database use

Personal genomic sequencing continues to drop in price and increase in accessibility. The inherent inability to truly anonymise such data, coupled with the wealth of information it provides on both individuals and families, distinguishes it from conventional data types such as fingerprints (identifiable but uninformative) or shopping habits (*Finnegan and Hall, 2017*). The drop in price and the use of technologies such as cloud storage have allowed wider use of DNA databases by different actors. While the vulnerability of cloud infrastructure is a concern, there is greater potential for misuse by states and law enforcement in the name of security. This has been seen in efforts to target Muslim Uighurs in China via blood samples (*Wee, 2019*), and in a consumer genetics database allowing the Federal Bureau of Investigation in the US to compare genetic data from crime scenes to a database of over two million profiles without customer consent (*Haag, 2019*). The potential to accrue and analyse vast amounts of genomic information raises concerns over privacy, especially mass surveillance (*Solove, 2011*); the potential expansion of state surveillance powers necessitates dialogue and policy intervention domestically and internationally.

## Issues most relevant in 5–10 years

### Agricultural gene drives

Gene drives were initially proposed for the control of insect vectors for human diseases (*Gantz et al., 2015*; *Neve, 2018*), but recent work suggest that they could provide major economic benefits to the agricultural sector (*Collins, 2018*; *Neve, 2018*). However, while there is potential for gene drives to eliminate or suppress pest species, their widespread uptake and use could lead to problems in their application and governance (*Evans and Palmer, 2018*). One concern is that commercial interests will seek to maintain sales of agrochemicals by configuring gene drives to reduce chemical resistance in target pest insects and weeds as opposed to causing sterility in those species. A second concern is that unilateral deployment of gene drives may cause rapid and unintended ecosystem perturbations without proper oversight or recall. There have also been questions around their control and the lack of public consultation (or participation) regarding their release, as well as legal implications if populations are eliminated within,

or new gene configurations are carried to, native locations (*Montenegro de Wit, 2019*). Efforts are already underway to counter, control and even reverse the undesired effects of genome editing, including DARPA's Safe Genes program (*Wegrzyn, 2019*). Policy-makers will need to be vigilant to more problematic applications as agricultural gene drives become more prevalent.

### Neuronal probes expanding new sensory capabilities

New research into creating probes that mimic neurons could enable novel medicinal and enhancement applications such as the creation of new sensory capabilities. Traditionally, neuronal probes have both structural and mechanical dissimilarities from their neuron targets, leading to neuro-inflammatory responses. However, it is now possible to fabricate neuron-like electronic probes (with widths similar to those of neurons) and unobtrusively fuse them with live neurons (*Yang et al., 2019b*). Potentially, the technology could be used to add new sensory capabilities by implanting neuronal probe arrays as a visual cortical prosthesis system. However, such biomimetic sensory probes could introduce unintended vulnerabilities, from a risk of malicious attack via the internet to possible mass monitoring of implanted civilians by law enforcement (*Yetisen, 2018*).

### Distributed pharmaceutical development and manufacturing

Outsourcing and increasingly lower barriers to access in bioengineering are allowing for greater localisation and geographical distribution of the manufacturing and development of pharmaceuticals. Bioengineering offers the capacity to create pharmaceutical compounds or their precursors by genetically modifying organisms to produce them. The prospect of non-traditional pharmaceutical manufacture has gained some traction, but with few tangible results. Barriers to distributed pharmaceutical manufacturing becoming broadly adopted include the scale of production required for individual or community use; meeting appropriate safety standards for manufacturing and administration; and interfacing with drug approval pathways. Efforts in non-traditional pharma, such as The Open Insulin Project (*Gallegos et al., 2018*), are rising in profile and will likely continue, whether individual projects are successful or not. This is supported by the Open Pharma movement which seeks to empower innovation through open-access

research and development (*Munos, 2010*; *Gassmann et al., 2018*; *Open Source Pharma, 2020*). That itself may shape regulatory frameworks, and may provide new open or distributed models for drug manufacturing. However, in the absence of appropriate norms or regulations (*Blum, 2010*), it may also lead to the manufacturing, at scale, of drugs that are not vetted for safety, or administered under appropriate clinical guidance (*Coleman and Zilinskas, 2010*).

### Genetically engineered phage therapy

The World Health Organization (WHO) recently reported a worrying lack of new antibiotics to address the dangerous trends of rising resistance to existing antibiotics (*WHO, 2015*), and antimicrobial resistance has been identified as a potential global catastrophic risk. Phage therapy has recently seen a renaissance as a potential alternative to antibiotic treatment. In particular, the ability to rapidly engineer phage sequences and phage cocktails opens up the prospect of personalised treatments for tackling genetically-diverse infections and overcoming problems of antimicrobial resistance (*Schmidt, 2019*). The technical advances observed in the medical application of phage therapy will also have an impact on other uses of phages as delivery systems in biotechnology. Efforts have also been significantly buoyed by the development of easier methods for engineering phages to combat the inevitable evolution of phage resistance in bacteria (*Pires et al., 2016*). However, barriers to widespread commercial use persist, including high costs, instability of the medication, the need to type the infection (instead of giving a broad-spectrum pill) and immunogenicity. This makes it more likely for phage therapy to be used as a last resort once other treatments have failed.

### Human genomics converging with computing technologies

Human genomics is increasingly incorporating technologies such as blockchain, cloud computing and machine learning. Firms such as Amazon and Google offer cloud computing-based storage and data analytics services for the petabytes of genetic data stored online, while companies such as Encrypgen and Nebula use blockchain in systems that reward individuals for sharing their genetic data. Artificial intelligence and machine learning are enabling deep analysis of thousands of molecules with potential to become future

drugs (*Japsen, 2016*), as well as human genomic data (*iCarbonX, 2018*). Most recently, deep learning used molecular structure to predict the efficacy of antibiotic candidates (*Stokes et al., 2020*). Some uses of these technologies could help address current privacy concerns. This includes the use of blockchain as well as 'secret sharing' techniques in which sensitive information is divided across multiple servers (*Cho et al., 2018*). However, as they are applied to human genomic data in increasingly powerful and connected ways, additional ethical issues will arise. Enlivened and global discussion on how best to handle societal implications will become necessary (*Yakubu et al., 2018*).

### Microbiome engineering in agriculture

Progress on microbiome engineering and genomic sequencing could allow for beneficial new applications in agriculture, but also risks. Microbiome engineering and the development of synthetic microbiomes offer wide-ranging uses for mammalian health as well as plant and animal productivity, soil health and disease management. A bottom-up approach to microbiome engineering aims to predictably alter microbiome properties and design functions for agricultural and therapeutic applications. Microbiome engineering strategies could provide alternatives to the use of antibiotics for livestock management (*Broaders et al., 2013*). These approaches offer the potential for innovative, sustainable pathways for plant disease suppression by engineering the microbiomes indigenous to agricultural soils (*Foo et al., 2017*). Advances in genome sequencing, metagenomics and synthetic biology have already provided a theoretical framework for constructing synthetic microbiomes with novel functionalities. New methods, such as in situ mammalian gut microbiome engineering, could help to overcome existing limitations and offer new capabilities for the future (*Ronda et al., 2019*). These new methods and advances can support better design of microbiome modulation strategies in mammalian health and agricultural productivity. Yet, the engineering of agricultural microbiomes on a large scale could also create vulnerabilities towards malicious intervention.

### Phytoremediation of contaminated soils

Research in phytoremediation is leading to the creation of engineered plants that could help recuperate contaminated soils, but further field trials are needed along with discussions about their introduction to and implications for the environment. Certain plant species have natural mechanisms that enable both uptake and tolerance of natural and anthropogenic inorganic pollutants. Identifying, expressing and potentially engineering these traits is receiving increased research interest. Preliminary work on transgenic plants in the lab by overexpression of metal ligands, transporters and specific enzymes has led to successful phytoextractions of pollutants including explosives and heavy metals. However, few experiments have been conducted in the field on contaminated soils (*Fasani et al., 2018*), where toxicity of various pollutants and the impact of various environmental factors on the plant-microbiome interaction has limited the success of phytoremediation to date. Realising biotechnological phytoremediation will depend on a number of factors: a more robust systemic understanding of plant-microbiome interactions with pollutants (*Basu et al., 2018*); the survivability of these engineered organisms in the environment; understanding and controlling environmental impacts; and robust societal discussion and carefully designed regulatory regimes.

### Production of edible vaccines in plants

Plants offer a scalable low-cost platform for recombinant vaccine production (*Merlin et al., 2017*). The introduction of the oral polio vaccine in the 1960s led to huge interest in developing vaccines that can be delivered without the need for injection. Given that plants are widely consumed, they offer an attractive means of vaccine delivery. Plant-expressed antibodies can protect against tooth decay. Similarly, expression of norovirus-like particles in transgenic potatoes could raise antibodies against the virus when the material is consumed (*Tacket et al., 2000*). Plant-produced vaccines have also been developed for some animal diseases (*Marsian et al., 2019*). Oral delivery with minimal processing has the potential to reduce requirements for extensive frameworks for production, purification, sterilisation, packaging and distribution. A major challenge is the need for improvement of the chemical and physical stability of vaccines during transit through the gut in order to ensure efficacy (*Berardi et al., 2018*). Also, commercialisation may be difficult under current regulatory regimes (*Merlin et al., 2017*). Moreover, if production is scaled up beyond contained greenhouses, this will require the deliberate environmental (field) release of plants engineered to contain vaccines.

### The rise of personalised medicine such as cell therapies

There is an accelerating trend towards the development and approval of personalised therapeutics. These are medical treatments that are tailored towards individuals, accounting for their likely response based on genomic and epigenetic data. In the US in 2018, 42% of all new drug approvals by the Food and Drugs Administration concerned these treatments (*PMC, 2019*). However, significant challenges stand in the way of developing and deploying personalised medicine and cell therapies. These includes issues of delivery logistics and cost. The key factor to clinical adoption of personalised medicine is the value recognition by all healthcare stakeholders. Most personalised medicines are genetically guided interventions that address relatively small subsets of patients with rare genetic mutations. The treatment approaches are sometimes costlier due to their increased sophistication and lower demand. Once these barriers are overcome there will be some potential problems that will need to be mitigated via policy. One is ensuring equitable access. Reimbursement from third-party payers such as health insurance companies is also likely to become an issue for targeted treatments (*Bilkey et al., 2019*; *Genetics Home Reference, 2019*). Public health policy must adapt to this new frontier of healthcare while addressing its potentially detrimental effects on equality of healthcare access and treatment.

## The issues most relevant in 10 years or more

### Bio-based production of materials

Biological engineering and production methods facilitate the transformation of renewable plant feedstocks and microorganisms into substitutes for a wide range of existing and new materials, including plastics and other materials that are produced from fossil fuels (*European Commission, 2017*). These developments are being driven by increasing government, private and civil society efforts to decarbonise economies. New opportunities may be created for small, bio-based production facilities and clean bio-refineries to be located close to the markets for these materials, potentially replacing much of the petrochemical sector, and there are potential roles for rural areas in growing bio-based feedstocks. While bio-based production promises to be more sustainable than existing methods, attention is still required in addressing specific impacts on feedstocks, energy, water and other environmental and societal factors (*Matthews et al., 2019*). This is accompanied by technical barriers in product processing. While some bio-based materials are already on the market, significant private investment and supportive public policy frameworks (including but not limited to carbon pricing, as well as more speculative nitrogen pricing) will be required over the next decade and beyond to accelerate the widespread worldwide transition to these materials (*HM Government, 2018*).

### Live plant dispensers of chemical signals

Plants emit volatile signals that can activate defence responses in other nearby plants. The concept of using GM plants to deliver these signals has made practical progress in recent years. These genetically modified plants are intended to be helpers that protect surrounding conventional crops that are cultivated for consumption. Field trials have evaluated the potential of transgenic wheat to repel different pests and virus vectors (*Bruce et al., 2015*). Despite excellent results in the lab, in planta synthesis of the alarm pheromone failed to reduce aphid numbers. Other studies have demonstrated the feasibility of making insect sex pheromones to trap male insects (*Ding et al., 2014*). Further finessing of the pheromone blend may be enabled by synthetic biology. This could open up the possibility of using plants as chemical-producing green factories, or field-based disruptors and dispersers of insect pests. Unlike current GM solutions for protection from insect herbivory, the use of pheromones is a non-lethal and less-persistent intervention, and chemically-manufactured pheromones have been in use for many years. Questions remain as to whether the broader adoption of pheromones will simply displace pests to unprotected crops.

### Malicious use of advanced neurochemistry

Agents that could attack the central nervous system were investigated during the Cold War but lack of knowledge only permitted the development of sedating agents. Concerns over such agents and manipulations continues (*Ward, 2019*), but could be empowered through advances in neuroscience and other fields. A driving force in these advances is significant government interest and investments, including an investment of almost $1bn by the US government in the Brain Initiative (*NIH, 2019*).

Resulting drugs and nootropics offer health benefits, but could also be maliciously used (*Nixdorff et al., 2018*). Governments could use neuro-chemicals to make a populace more subservient. Advanced applications in undeclared biological warfare could include fostering emotional resentment in a targeted population. These drugs could be appealing to governments around the world as a tool for counter-insurgency or non-lethal law enforcement. The use of these new chemicals for law enforcement and in non-traditional conflicts may greatly erode the norms against chemical agent use on the battlefield, threatening the Chemical Weapons Convention in the long term.

### Enhancing carbon sequestration

Metabolic engineering manipulates cells to produce target molecules by optimising endogenous metabolic pathways or by reconstructing these pathways in alternative species. 'Next level' metabolic engineering aims to design metabolic networks de novo, thus bypassing the bottlenecks and inefficiencies of evolution (*Erb, 2019*). Thus far, experimental success is lacking. However, recent research in photosynthesis may be promising, and examples include engineering a new molecule to perform a designed synthetic photorespiration bypass (*Trudeau et al., 2018*) and developing an optimised carbon dioxide fixation pathway using enzymes from bacteria, archaea, plants and humans (*Schwander et al., 2016*). Other methods have included laboratory evolution of a bacterium able to use $CO_2$ for growth (*Gleizer et al., 2019*). These approaches hold potential for more efficient carbon sequestration and biomass production, as well as for advancing the development of photovoltaics (the production of electricity from light) and light-sustained biomanufacturing. Yet, such developments remain speculative. There are still significant technical challenges to overcome, and a long path to widespread commercial deployment. Moreover, the field will need to engage with its socio-political, ethical and environmental dimensions.

### Porcine bioengineered replacement organs

Pigs represent a promising candidate species for production of human-compatible replacement organs for xenotransplantation. A recent advance in porcine genome editing using CRISPR/Cas9 addresses one of the key scientific challenges: successful inactivation of porcine endogenous retroviruses, which otherwise pose a risk of cross-species transmission (*Niu et al., 2017*). Such advances hold promise as one technological way to address the global shortage of transplant organs. Over 6,500 patients died while on waiting lists in the USA alone in 2017 (*UNOS, 2019*). Several challenges remain, including engineering sufficient immune compatibility in the organs for successful human transplantation, and determining the expected lifespan of the porcine organs in humans. There are differing views over the acceptability of porcine xenotransplantation within major religions, such as Islam and Judaism (*Nuffield Council on Bioethics, 1996*). Before commercial development, consideration must be given to questions surrounding the ethics of using animals for transplantation, cost and access, and using a technical solution for an essentially social problem that could be addressed through other approaches, such as opt-out organ donation schemes.

### The governance of cognitive enhancement

Cognitive enhancement is already a widely embraced idea throughout society – caffeine is the most widely consumed drug on earth. Novel methods of cognitive enhancement such as nootropics, wakefulness enhancers, or the potential to directly modulate brain function through implants or biotechnology are emerging. Uptake of these is being driven by both a productivity-focused culture, commercial opportunities and increased understanding of neurochemistry. Although some cognitive enhancers require prescriptions, others only have to meet basic safety guidelines and are available to purchase online. While numerous trials have supported the safety of most nootropics and wakefulness enhancers, there are few long-term longitudinal studies (*Fond et al., 2015*). A large section of those who have embraced cognitive enhancement – the 'do-it-yourself' experimenters – may also be ignored by the research community. Lax regulation around safety standards for these products and tools has led to calls to tighten regulatory loopholes, and for academic researchers to partner with and include communities in research on cognitive enhancers (*Wexler, 2017*). Regulatory frameworks are necessary to both minimise risks and gather long-term safety data from end-users, as well as to provide health and safety guidance for international trade of cognitive enhancing drugs and devices (*Maslen et al., 2015*).

## Discussion

### Emergent themes

Seven underlying themes emerged from the workshop discussion: 1) political economy and funding; 2) ethical and regulatory frameworks; 3) climate change; 4) transitioning from lab to field; 5) inequalities; 6) technological convergence; and 7) misuse of technology. None of these were judged precise enough to qualify as horizon-scanning items, although some sub-components were. These themes represent underlying commonalities and drivers across issues.

First, participants expressed concern about the political economy of bioengineering (that is, how political and economic institutions influence bioengineering, including the role of regulation and politics) and, related to this, about funding. These concerns centred around a view that research funded by the military, industry or philanthropy was less accountable than civilian government-funded research and could create real or perceived conflicts of interest (see, for example, *Licurse et al., 2010*).

Second, a recurring theme across several issues was the need for ethics and better regulatory frameworks to manage the problems expected to emerge from technologies on the horizon. This was true for most issues highlighted in the scan, ranging from carbon sequestration to bioengineered replacement organs. This underscores the need for greater engagement between ethicists, social scientists, policy-makers and the cutting-edge of bioengineering.

Third, climate change is likely to be a critical driver of bioengineering in the future. Our list includes an application to both adaptation (crops for changing climates) and negative emissions (sequestration). Others, such as live plant dispensers, could be boosted in relevance as a way to enhance agricultural productivity in the face of detrimental climate impacts. Progress in climate policies will shape the development and demand of bioengineering technologies. Climate change impacts will also create new problems that could be addressed through bioengineering and policy. This includes changes in the range of vector-borne diseases, such as the expansion of tropical infectious diseases.

A fourth theme is that of transitioning from lab to field. The deliberate release of a new bioengineering product into the environment entails risks in both practice and perception. Concerns over the unintended consequences of environmental release have hindered the deployment of GMOs and are now prominent in discussions around gene drives (*Evans et al., 2019*). Such concerns also factored into many of the issues we have identified, most notably edible vaccines and live plant dispensers. Further development of bioengineered products will require appropriate regulation. Additionally, the necessary social, environmental and human health risk assessments need to take place to transition bioengineering from the lab into the wider world.

A fifth theme is the potential for bioengineering to exacerbate existing inequalities in wealth and health. This factored into several issues including the rise of personalised medicine, replacement organs, and the regulation of cognitive enhancement. In contrast, distributed pharmaceutical development and manufacturing was an emerging area fuelled in part by the desire to deliver more equitable, cheap and accessible medicine. Ensuring that the benefits of bioengineering are spread fairly and widely will be a defining feature of future debates. Enhancements also come with risks, especially at the earliest stages. Many of these are expected to be borne by unwilling or uninformed recipients (as in the case of the CRISPR twins) before being marketed to the wealthy. These problems of inequality also highlight the need for horizon-scanning efforts to make efforts to include representatives from more oppressed and marginalised groups.

The sixth theme is that the convergence of different technologies will be crucial in the future development of bioengineering. Many of the issues in this horizon scan are driven by progress in adjacent fields. Both neuronal probes and malicious uses of neurochemistry will be enabled by progress in neuroscience, and the overlap of human genomics with computing technologies brings both opportunities and threats. As automation and measurement, neuroscience, chemistry and artificial intelligence continue, they will shape both what is possible and what is pursued in bioengineering. This poses a challenge for regulators, who may need to think about policy that cuts across bioengineering into other areas, such as cybersecurity. It also highlights a need for continued horizon scanning and foresight exercises to engage a broad range of technological expertise so that key points of intersection and convergence are not overlooked.

Last, our scan highlights ongoing concerns around the misuse of technology by state or non-state actors. Examples included various bioweapons and the misuse of DNA databases.

**Table 2.** Overview of the bioengineering horizon scan 2017.

Summary of the 20 issues identified in 2017; issues are grouped according to likely timeline for realisation.

| <5Years | 5–10 Years | >10 Years |
|---|---|---|
| Artificial photosynthesis and carbon capture for producing biofuels | Regenerative medicine: 3D printing body parts and tissue engineering | New makers disrupt pharmaceutical makers |
| Enhanced photosynthesis for agricultural productivity | Microbiome-based therapies | Platform technologies to address emerging disease pandemics |
| New approaches to synthetic gene drives | Producing vaccines and human therapeutics in plants | Challenges to Taxonomy-Based description and management of biological risk |
| Human genome editing | Manufacturing illegal drugs using engineered organisms | Shifting ownership models in biotechnology |
| Accelerating defense agency research in biological engineering | Reassigning codons as genetic firewalls | Securing the critical infrastructure needed to deliver the bioeconomy |
|  | Rise of automated tools for biological design, test and optimisation |  |
|  | Biology as information science: impacts on global governance |  |
|  | Intersection of information security and bio-automation |  |
|  | Effects of the Nagoya Protocol on biological engineering |  |
|  | Corporate espionage and biocrime |  |

The 2017 scan noted themes of equality, bioinformatics and regulation, all of which feature prominently in the 2020 scan (see *Table 2* for a summary of the previous scan). The 2017 exercise discussed the intersection between biotechnology and information and digital technologies. Technological convergence also features in the present scan, but with a broader scope encompassing neuroscience (adding new sensory capabilities) and neurochemistry (malicious uses of advanced neurochemistry) as well as other fields. Both scans featured a strong emphasis on the potential for bioengineering to amplify or alleviate inequalities. In the 2017 scan this included the potential for human genomics to create new 'sociogenetic' classes, while differences in healthcare and access to cognitive enhancement were the flagship issues in this 2020 scan. The thematic convergence between the two scans demonstrates that many of the underlying trends in bioengineering include important structural issues involving ethics and regulation. These will likely influence the field for years to come. There were also several differences in themes, including the greater importance of climate change

and political economy in the 2020 exercise. This reflects the significant deviation in issues between the two studies.

Some issues from 2017 also appear in the 2020 exercise in a slightly altered form: concerns about the military use of bioengineering are now more specific (for example, 'Malicious use of advanced neurochemistry'), and there are new concerns about the misuse of DNA databases.

Both scans also focussed on different methods for the production of replacement organs. The 2017 exercise identified 3D printing cells on organ-shaped scaffolds, while the 2020 exercise examined the potential for porcine genome editing to allow for xenotransplantation. Finally, both scans assessed the issue of pharmaceutical manufacturing becoming increasingly distributed. The 2017 exercise focused on start-up entrepreneurs and biohacking communities, whereas the 2020 exercise took a broader look at the possibility of decentralisation.

The differences between the scans are likely due to three reasons. First, we used a wider definition of bioengineering which encompassed issues such as biomechanical implants. Two of

the issues identified in this scan would not have been covered by the 2017 definition: neuronal probes expanding new sensory capabilities and the governance of cognitive enhancement. Second, half of the participants (19/38) were not involved in the 2017 scan; the new participants were also more geographically diverse (see Methods), and included a higher proportion of social scientists. Third, there have been significant changes in research and the world at large. For example, all the research underpinning the issue of neuronal probes has occurred in the last three years. Similarly, recent research in climate change has highlighted the continued increase in emissions and warming (*Friedlingstein et al., 2019*), and that tipping points are more probable than previously expected (*Steffen et al., 2018*; *Lenton et al., 2019*).

### *Limitations and ways forward*

While useful, horizon scanning has its limits. Critiques have suggested that the Delphi technique (of which the IDEA protocol is a relatively recent evolution) can give unjustified confidence in results that are essentially the subjective judgements of experts (*Sackman, 1975*). However, in the absence of data, expert elicitation is warranted, and structured approaches such as Delphi and the IDEA protocol have been found to improve group judgement and outperform other forecasting methods, such as prediction markets (*Hanea et al., 2017*). While it is difficult to evaluate the efficacy of the Delphi technique due to inconsistencies in its application (*de Loë et al., 2016*), those that do exist are promising. A review of a long-term Delphi in predicting developments in the health sector found that results were accurate in 14/18 identified issues (*Parente and Anderson-Parente, 2011*). The method continues to show significant utility in both accurately sighting emerging developments and exploring the implications of potential issues on the horizon.

We acknowledge that the issues identified in this horizon scan are ultimately representative of the participants involved. While the 2020 scan is an improvement on previous efforts in terms of diversity, the majority of respondents were still from a developed economy background. The scan did capture a large cross-section of academic sub-fields in bioengineering, but underrepresented industry, communities and policymakers. Moreover, we achieved a rough gender balance with 21 male participants (55%) and 17 female participants (45%). We intend to make the process increasingly global and diverse

under future triennial iterations, and by clearly describing the methods used, have made the process open for uptake by others.

Future pathways for forecasting bioengineering issues are manifold. Further updates of this scan could be paired with systematic reviews of their accuracy and efficacy, as well as deeper dives into the issues that have been identified. Extensions of the horizon-scanning process could include: focusing on specific areas of bioengineering, such as catastrophic risks; incorporating decision-support tools such as fault-trees; examining the development of bioengineering issues in tandem with overlapping technological areas such as artificial intelligence; and producing a policy-focused scan which involves greater engagement with regulators.

## Methods

Our study made use of the Investigate Discuss Estimate Aggregate (IDEA) protocol. In this process, participants were asked to investigate and submit candidate issues, privately and anonymously score the gathered issues, and discuss their thinking with others. They then provided a second score which was mathematically aggregated (*Hanea et al., 2018a*). The element of discussion is powerful, as the sharing of information between participants has been shown to improve the accuracy of Delphi-style forecasts (*Hanea et al., 2018b*). The IDEA protocol has also performed well relative to prediction markets in early studies (*Hanea et al., 2017*). Despite being a relatively recent evolution of the Delphi technique, the IDEA protocol has already been successfully applied to a range of areas including natural resource management (*Hemming et al., 2018*) and assessing pollinator abundance in response to environmental pressures (*Barons et al., 2018*). Aside from seeking a shared understanding of terms and reducing linguistic ambiguity, consensus is not sought during discussion and scores are kept anonymous during both rounds. This is done to avoid undesirable group dynamics and peer pressure distorting individual judgements. Our use of the IDEA protocol can be split into three phases: i) recruitment and issue gathering; ii) initial scoring; and iii) workshop preparation, deliberation and re-scoring.

### *Phase one: recruitment and issue gathering*

Our study drew on a group of 38 participants from six continents. Participants came from countries including the UK, US, Canada,

Australia, Germany, Croatia, Thailand, France, Chile, Peru, Switzerland, Malaysia, Zambia and Pakistan. Recruitment was done via a panel of six initial experts (EP, PM, SÓhÉ, CR-R, CR, LS and BW). The panel aimed to ensure a balance across areas such as plant sciences, medicine, bioindustry and biosecurity. They also sought to have a mix of approximately half new participants and half participants from the 2017 exercise scan. Selected bioengineering scholars and practitioners were asked to submit two to five issues each. Our initial request was for issues that were 'novel, plausible and high-impact'. We asked participants to provide issues that were at a specific level of granularity. As with the previous scan we asked participants not to focus on a general topic, such as 'gain of function' research, nor on multiple topics simultaneously. Instead they were guided to focus on one area within a general topic and its implications, such as an emerging regulatory change for GMOs. After duplicates were merged, a long list of 83 issues was generated from the initial submissions. This included 10 merged issues.

### Phase two: scoring

Participants were asked to vote on the 'suitability' of these issues. This involved assigning a score of 0–1,000 to each of the issues. Participants were asked to ensure that each score was unique (no identical scores within a given score-sheet). The suitability scores reflected a combination of plausibility, novelty and impact. Novelty was also captured by respondents noting whether they had heard of the issue previously (through a 'yes/no' response). We then calculated the percentage of participants who had heard of each issue. These novelty scores were published alongside all issues in the short list. This was conducted by sending the participants both the long list of issues, along with a template score-sheet and instructions. At this stage participants were reminded that "our aim is to identify plausible, novel bioengineering-related issues with important future implications for society that are not too broad or already well known'. They were given approximately three weeks to complete their scoring. All anonymised score-sheets are provided in *Supplementary file 1*; this file also includes the z-scores of the top 20 issues identified in the 2017 scan. Participants were also able to provide comments on the different issues on the voting sheet. These critiques led to a further eight issues being merged into four. Comments were kept to stimulate future discussion. We calculated the z-scores for each

participant's issues scores. Z-scores are created by subtracting the mean and dividing by the standard deviation for each issue against the participant's set. This ensures that variations in the range of participants' scoring is accounted for. We then ranked the average z-scores across the issues and selected the highest ranked 41 (approximately cutting the long list in half).

We discussed two potential reforms on the previous scoring approach: breaking scoring down across the three criteria, and including uncertainty estimates. We decided against both potential reforms. Experts are poor at estimating their own uncertainty and this could incentivise overconfidence. We decided that greater disaggregation in voting was likely to impose a greater burden on participants while providing little additional benefit. Moreover, keeping the protocol similar to the 2017 scan was desirable for comparison.

One amendment was made to the previous horizon-scanning methodology: the introduction of 'devil's advocates' into the process. *Goodwin and Wright, 2010* have noted that most forecasting methods are inadequate for identifying high-impact, low-probability events (some times called 'black swan events'). However, the Delphi technique can be better suited to the task if it includes devil's advocates who can advocate for less likely but significant issues. We empowered two individuals during the first phase of the process to propose more speculative and transformative issues. Two different participants were then asked during the third phase (workshop deliberation) to provide more critical inputs and actively push against the prevailing, dominant view during discussions. In each case their designation was not revealed to the group.

The devil's advocates appear to have been a useful addition and were disproportionately successful in suggesting issues. Six of the nine issues they proposed in the first round made it through to the short list, and four of the six issues they proposed in the second round made it through to the final list of 20; with 38 participants, we would expect approximately only one issue for every second participant to make it through to the final list. 68% of participants had heard of the issues proposed by the devil's advocates, making these issues moderately more novel than the rest. Overall, an average of 70% of participants had heard of each issue. The level of novelty of the issues suggested by devil's advocates is partly skewed by two more well-known issues which both scored 82.35%. When both of these issues were excluded, the devil's

**Table 3.** A comparative analysis of the groups involved with phases one and two, and phase three (the workshop).

| Characteristics | Phases one and two | Phase three (workshop) |
|---|---|---|
| Sample Size | 38 | 25 |
| Gender Balance | 21 male participants (55%) and 17 female participants (45%) | 13 females (52%) and 12 males (48%). |
| Geographical Coverage | 13 countries (UK, US, Canada, Australia, Germany, Croatia, Thailand, France, Chile, Peru, Switzerland, Malaysia, Zambia and Pakistan) | 10 countries (UK, US, Canada, Australia, Germany, Croatia, Thailand, France, Chile, Switzerland and Pakistan) |
| Disciplinary Distribution | 15 participants from humanities and social sciences (39%) and 23 from natural sciences (61%) | 9 participants from humanities and social sciences (36%) and 16 from natural sciences (64%) |

advocates suggestions were significantly more novel at an average of 61%.

### Phase three: workshop preparation, deliberation and re-scoring

The 41 issues with the highest scores were kept as a part of a shortlist. These were sent back to participants on the 13th of September 2019. Participants were assigned 'cynic' roles for each issue. This involved doing deeper background research into the topic. Each issue had at least two cynics, ensuring that at least three participants (the cynics and proposer) had an in-depth knowledge of the area. The workshop was held in Cambridge on the 9th of October 2019 with 25 participants; 13 could not attend due to other obligations. This resulted in a group with approximately the same characteristics as the group that was involved in he first two phases. The characteristics of both groups are compared in *Table 3*. Overall, the gender balance was maintained (although the slight skew was reversed towards female participants), the disciplinary split between social and physical scientists was approximately the same, and the geographical coverage became less balanced due to the loss of participants from Peru, Zambia and Malaysia.

These discussions were overseen by an experienced facilitator (WJS, with LK and AR acting as scribes) and followed a deliberate structure. Each issue was discussed for approximately ten minutes before being voted on anonymously. During discussions, proposers of the issue were asked not to speak until at least three other respondents had contributed. This was done to avoid biasing the conversation and allowing the cynics time to provide an orientating, more neutral intervention. The standardised z-scores for each issue were calculated and ranked at the end of the workshop, resulting in a top 20 list. The decision to keep the list to 20 was made by consensus by the workshop group and was influenced by a significant difference between the

z-scores of the top and bottom 20 issues, but a much smaller spread of scores within the top 20. Participants were then given time to discuss the final list and whether any amendments were needed. The group was content with the spread of the final list and that it accurately reflected the deliberations and hence decided that no alterations were needed.

A comparison of the rankings of the top 20 issues after the first and second round of scoring can be found in *Supplementary file 2*. There was a noticeable difference between the two rankings. For example, 11 out of the final top 20 (55%) issues had been ranked outside of the top 20 during the first round of scoring. Indeed, four of the top five issues (80%) were outside the top 20 after the first round of scoring. This suggests that deliberation was effective in shifting participants' perspectives and scores. The novelty scores are also summarised in *Supplementary file 2*. The final list of issues had a slightly higher degree of novelty, but this was minor. The short list of issues resulted in an average score of 70.52% and a median of 73.53%. By contrast the final list had an average of 68.97% and a median of 67.75%.

### Acknowledgements

First and foremost we thank Clare Arnstein for her indispensable work in helping to organise the horizon-scanning process and workshop. We thank Andrew Balmer, George Church, Nick Matthews, Robert Meckin, Penny Polson, Fatemeh Salehi and Andrew Watkins for their involvement in submitting issues. This project was made possible through the support of a grant from Templeton World Charity Foundation. The opinions expressed in this publication are those of the authors and do not necessarily reflect the views of Templeton World Charity Foundation.

**Luke Kemp** is in the Centre for the Study of Existential Risk (CSER) and the Biosecurity Research Initiative at St

Catharine's College, University of Cambridge, Cambridge, United Kingdom
ltk27@cam.ac.uk
https://orcid.org/0000-0002-7447-4335

**Laura Adam** is in Ebiosec, Inc, Seattle, United States
https://orcid.org/0000-0002-4822-2695

**Christian R Boehm** is in the Centre for the Study of Existential Risk (CSER), University of Cambridge, United Kingdom
https://orcid.org/0000-0002-6633-7998

**Rainer Breitling** is in the Manchester Institute of Biotechnology, Faculty of Science and Bioengineering, University of Manchester, United Kingdom
https://orcid.org/0000-0001-7173-0922

**Rocco Casagrande** is in Gryphon Scientific, Takoma Park, United States

**Malcolm Dando** is in the Division of Peace Studies and International Development, University of Bradford, Bradford, United Kingdom

**Appolinaire Djikeng** is in the Centre for Tropical Livestock Genetics and Health, Royal (Dick) School of Veterinary Studies, University of Edinburgh, Edinburgh, United Kingdom
https://orcid.org/0000-0002-0955-171X

**Nicholas G Evans** is in the Department of Philosophy, University of Massachusetts Lowell, and at Rogue Bioethics, Lowell, United States
https://orcid.org/0000-0002-3330-0224

**Richard Hammond** is at Cambridge Consultants, Cambridge, United Kingdom

**Kelly Hills** is in Rogue Bioethics, Lowell, United States

**Lauren A Holt** is in the Centre for the Study of Existential Risk (CSER) and the Biosecurity Research Initiative at St Catharine's College, University of Cambridge, Cambridge, United Kingdom

**Todd Kuiken** is at the Genetic Engineering and Society Center, North Carolina State University, Raleigh, United States
https://orcid.org/0000-0001-7851-6232

**Alemka Markotić** is in the Medical School, University of Rijeka, the University Hospital for Infectious Diseases, and the Catholic University of Croatia, Croatia

**Piers Millett** is in the Future of Humanity Institute, University of Oxford, Oxford, UK, and in the iGem Foundation, Boston, United States

**Johnathan A Napier** is at Rothamsted Research, Harpenden, United Kingdom
https://orcid.org/0000-0003-3580-3607

**Cassidy Nelson** is in the Future of Humanity Institute, University of Oxford, Oxford, United Kingdom
https://orcid.org/0000-0003-0934-8313

**Seán S ÓhÉigeartaigh** is in the Centre for the Study of Existential Risk (CSER) and the Biosecurity Research Initiative at St Catharine's College, University of Cambridge, Cambridge, United Kingdom

**Anne Osbourn** is in the John Innes Research Centre, Norwich, United Kingdom

https://orcid.org/0000-0003-2195-5810

**Megan J Palmer** is in the Center for International Security and Cooperation (CSIAC) and the Department of Bioengineering, Stanford University, Stanford, United States
https://orcid.org/0000-0001-8310-2325

**Nicola J Patron** is in the Earlham Institute, Norwich, United Kingdom
https://orcid.org/0000-0002-8389-1851

**Edward Perello** is at Arkurity Ltd., London, United Kingdom

**Wibool Piyawattanametha** is in the Biomedical Engineering Department, Faculty of Engineering, King Mongkut's Institute of Technology Ladkrabang, Bangkok, Thailand, and in the Institute of Quantitative Health Sciences and Engineering, Michigan State University, East Lansing, United States
https://orcid.org/0000-0002-2228-8485

**Vanessa Restrepo-Schild** is in the Chemistry Research Laboratory, University of Oxford, Oxford, United Kingdom
https://orcid.org/0000-0002-3486-454X

**Clarissa Rios-Rojas** is in the Centre for the Study of Existential Risk (CSER), University of Cambridge, Cambridge, UK, and at Ekpa'Palek: Empowering Latin-American Young Professionals, Huacho, Peru
https://orcid.org/0000-0001-6544-4663

**Catherine Rhodes** is in the Centre for the Study of Existential Risk (CSER) and the Biosecurity Research Initiative at St Catharine's College, University of Cambridge, Cambridge, United Kingdom
https://orcid.org/0000-0002-7747-2597

**Anna Roessing** is in the Department of Politics, Languages and International Studies, University of Bath, Bath, United Kingdom
https://orcid.org/0000-0002-2888-3963

**Deborah Scott** is in Science, Technology & Innovation Studies at the School of Social and Political Sciences, University of Edinburgh, Edinburgh, United Kindom
https://orcid.org/0000-0002-5962-2002

**Philip Shapira** is in the Manchester Institute of Innovation Research, Alliance Manchester Business School and SYNBIOCHEM, University of Manchester, Manchester, United Kingdom, and the School of Public Policy, Georgia Institute of Technology, Atlanta, United States
https://orcid.org/0000-0003-2488-5985

**Christopher Simuntala** is in the National Biosafety Authority, Lusaka, Zambia

**Robert DJ Smith** is in Science, Technology & Innovation Studies, School of Social and Political Science, University of Edinburgh, Edinburgh, United Kindom
https://orcid.org/0000-0002-5814-6032

**Lalitha S Sundaram** is in the Centre for the Study of Existential Risk (CSER) and the Biosecurity Research Initiative at St Catharine's College, University of Cambridge, Cambridge, United Kingdom

https://orcid.org/0000-0002-9595-9753

**Eriko Takano** is in the Manchester Institute of Biotechnology, Faculty of Science and Bioengineering, University of Manchester, United Kingdom

https://orcid.org/0000-0002-6791-3256

**Gwyn Uttmark** is in the Department of Chemistry, Stanford University, Stanford, United States

https://orcid.org/0000-0001-8758-5256

**Bonnie C Wintle** is in the School of BioSciences, University of Melbourne, Melbourne, Australia

**Nadia B Zahra** is in the Department of Biotechnology, Qarshi University, Lahore, Pakistan

**William J Sutherland** is in the Biosecurity Research Initiative at St Catharine's College and the Department of Zoology, Cambridge University, Cambridge, United kingdom

w.sutherland@zoo.cam.ac.uk

*Author contributions:* Luke Kemp, Conceptualization, Data curation, Formal analysis, Investigation, Methodology, Writing - original draft, Writing - review and editing; Laura Adam, Christian R Boehm, Rainer Breitling, Rocco Casagrande, Malcolm Dando, Appolinaire Djikeng, Nicholas G Evans, Richard Hammond, Kelly Hills, Lauren A Holt, Todd Kuiken, Alemka Markotić, Johnathan A Napier, Cassidy Nelson, Anne Osbourn, Megan J Palmer, Nicola J Patron, Wibool Piyawattanametha, Vanessa Restrepo-Schild, Anna Roessing, Deborah Scott, Philip Shapira, Robert DJ Smith, Lalitha S Sundaram, Eriko Takano, Bonnie C Wintle, Nadia B Zahra, Investigation, Writing - original draft, Writing - review and editing; Piers Millett, Edward Perello, Clarissa Rios-Rojas, Investigation, Methodology, Writing - original draft, Writing - review and editing; Seán S ÓhÉigeartaigh, Funding acquisition, Investigation, Methodology, Writing - original draft, Writing - review and editing; Catherine Rhodes, Supervision, Funding acquisition, Investigation, Methodology, Writing - original draft, Project administration, Writing - review and editing; Christopher Simuntala, Gwyn Uttmark, Investigation, Writing - review and editing; William J Sutherland, Conceptualization, Formal analysis, Supervision, Funding acquisition, Investigation, Methodology, Writing - original draft, Project administration, Writing - review and editing

*Competing interests:* Nicholas G Evans: Has received funding in the past five years from the following organizations to conduct research and speak on issues related to this article: Greenwall Foundation; Max Planck Institute for Evolutionary Biology; Davis Foundation; National Science Foundation. Has participated in the International Genetically Engineered Machine (iGem) competition as a judge and a member of the Safety Committee several times over the past five years. Kelly Hills: Has participated in the International Genetically Engineered Machine (iGem) competition as a judge and a member of the Safety Committee several times over the past five years; has accepted travel funding from the Max Planck Institute for Evolutionary

Biology. Todd Kuiken: Holds a grant from the Open Philanthropy Foundation to study DIYbio, including gene drives; member of the IUCN task force on synthetic biology and gene drives; co-chair of the human practices committee of iGEM; member of the biosafety/security advisory board for MIT Broad Foundry; has been affiliated with the Genetic Biocontrol of Invasive Rodents consortium at NC State but has never received any salary or other compensation. Piers Millett: Co-founder of Biosecure Ltd. Johnathan A Napier: Listed as an inventor on multiple patents; most relevant patent is US 2019/0323022 A1 and family. Acted as a consultant to BASF (2014-2019). Anne Osbourn: Director of the Industrial Biotechnology Alliance, Norwich Research Park; trustee with the New Phytologist Trust. Megan J Palmer: Recipient of a grant and a gift from the Open Philanthropy Foundation to study biosecurity at Stanford university; volunteer director of the human practices committee of the international Genetically Engineered Machine (iGEM) Foundation; member of the board of directors of Revive & Restore. Edward Perello: Holds shares in Celixir Ltd and Arkurity Ltd; has received payment from Deep Science Ventures Ltd, the Nuclear Threat Initiative, Revive & Restore, Celixir Ltd, Arkurity Ltd, CSIRO, International Union for the Conservation of Nature and Biosecure. Deborah Scott, Robert DJ Smith: Involved in a consultancy for the ERA-CoBioTech programme (an ERA-Net network funded by Horizon 2020). The other authors declare that no competing interests exist.

*Ethics:* Human subjects: Contributors were informed and agreed with the intent to publish at the outset, and were aware that they could withdraw at any time. Initial contact via email with all contributors outlined the conditions of the exercise, details about the previous scan, the expectations of their involvement (such as the time required) and the intent to publish the results of the exercise with them acting as co-authors. Participants were all above the age of 18 and agreed to these terms via email. The decision to publish with eLife was reached by consensus by contributors. We followed the Key Principles of Ethical Research from the Humanities and Social Sciences Research Ethics Committee of the University of Cambridge. Board approval was not deemed necessary as the study did not involve minor, vulnerable groups, manipulation, or any research beyond the UK. All contributors were reimbursed for their costs in participating in the horizon-scanning workshop.

### Funding

| Funder | Grant reference number | Author |
|---|---|---|
| Templeton World Charity Foundation | TWCF0128 | Luke Kemp |

| | | |
|---|---|---|
| Biotechnology and Biological Sciences Research Council | BB/M017702/1 | Rainer Breitling Philip Shapira Eriko Takano |
| Engineering and Physical Sciences Research Council | EP/S01778X/1 | Philip Shapira Rainer Breitling Eriko Takano |
| European Research Council | ERC 616510 | Deborah Scott Robert D.J. Smith |
| Biotechnology and Biological Sciences Research Council | BB/R021554/1 | Nicola J Patron |
| Biotechnology and Biological Sciences Research Council | BB/P010490/1 | Nicola J Patron |
| Biotechnology and Biological Sciences Research Council | BB/S020853/1 | Nicola J Patron |

The funders had no role in study design, data collection and interpretation, or the decision to submit the work for publication.

**Decision letter and Author response**
Decision letter https://doi.org/10.7554/eLife.54489.sa1
Author response https://doi.org/10.7554/eLife.54489.sa2

## Additional files

### Supplementary files

• Supplementary file 1. Scoring of the issues in the 2017 and 2020 scans. Raw and standardised scoring of the issues in the 2020 scan after the first and second rounds of the exercise, novelty scores of the issues in the 2020 scan, and standardised z-scores for the final 20 issues of the 2017 scan.

• Supplementary file 2. Ranking of the 20 issues after the first and second rounds of scoring, and summary of the novelty scores. Some of the top 20 issues had their titles changed after the workshop. In these cases, we have listed the final title in brackets next to the original title.

• Transparent reporting form

### Data availability

All data generated or analysed during this study are included in the manuscript and supporting files. We have also attached the anonymised scoresheets as supplementary data. These are also available via the Open Science Framework- https://osf.io/ej8p2/.

The following dataset was generated:

| Author(s) | Year | Dataset URL | Database and Identifier |
|---|---|---|---|
| Kemp L, Adam L, Boehm CR, Breitling R, Casagrande R, Dando M, Djikeng A, Evans NG, Hammond R, Hills K, Holt LA, Kuiken T, Markotić A, Millett P, Napier JA, Nelson C, ÓhÉigeartaigh SS, Osbourn A, Palmer M, Patron NJ, Perello E, Piyawattanametha W, Restrepo-Schild V, Rios-Rojas C, Rhodes C, Roessing A, Scott D, Shapira P, Simuntala C, Smith RD, Sundaram LS, Takano E, Uttmark G, Wintle B, Zahra NB, Sutherland WJ | 2020 | https://osf.io/ej8p2/ | Open Science Framework, ej8p2 |

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
