## [Decision Letter]

Thank you for submitting your article "Bioengineering Horizon Scan 2020" for consideration by *eLife*. Your article has been reviewed by three peer reviewers, and the evaluation has been overseen by two editors from the *eLife* Features team (Helena Pérez Valle and Peter Rodgers). The following individuals involved in review of your submission have agreed to reveal their identity: Ariel B Lindner (Reviewer #3).

Summary:

Kemp et al. present a timely and well-written qualitative analysis of short (<5 years) - to long- (>10) term emerging trends in bioengineering and succinctly describe hurdles as well as potential positive and negative outcomes.

The manuscript is an update of a similar exercise carried out in 2017. The augmented Delphi process used to drive a consensus among self-selected experts is well-described, and its main deficiency (the 'loud voice' effect) was addressed by limiting proposers' contributions to the discussion. Overall, the manuscript is easily accessible for general scientific audience and comprehensible for policy makers and in large portions readable by the general public. However, the manuscript would benefit from addressing a number of issues - see below.

Essential revisions:

1. Restructure the Introduction so that it starts with a discussion of why the scan is needed, why Bioengineering in particular is of interest, and what gap the work is trying to fill. Limit the discussion of the methods to the end of the Introduction.

2. Table 1 and the review of the 20 issues are results, so a Results section should be included with these elements.

3. Give a more detailed (but still brief) explanation about how the scoring of the issues works in the Introduction, as well as a more detailed explanation in the Materials and Methods. Also include why 20 was chosen as a cutoff for the number of issues finally reported. Please provide (either in the article or as supplementary data) a list of the 83 issues initially proposed and the z-scores calculated.

4. For each of the sections describing one of the issues (currently in the Introduction, to be moved to a Results section, see above), clearly identify in each what the emerging issue is, why it's a (potential) issue (positive or negative), what the drivers of the issue are, and the current state of the issue to provide a stronger takeaway message about why each issue is important.

5. On the comparison between the 2017 and 2020 scan:

i) Expand on the differences between the 2017 and the 2020 scan, including discussing whether any of the emerging issues of 2017 have come into effect; or why the experts may have switched focus (including whether expert selection may have biased this change; or whether any disruptive influences or drivers may have changed experts' focus).

ii) Map the final issues from the 2017 scan horizon to the full list of 83 issues of the 2020 scan horizon using the Z score matrix, rather than sharing the 2017 issues as a table. If the table is not openly accessible in the previous publication, include it here as supplementary data.

iii) If possible, designate which new issues on the 2020 list are due to a wider scope definition of bioengineering, otherwise the comparison between the 2017 and the 2020 scan horizons can be biased.

6. Report the gender/geographical/disciplinary bias in the 2nd as compared to the 1st experts' panel, since the 1st panel had 38 participants while the 2nd panel had 26.

7. Share statistics regarding the 'Novelty' yes/no response, and briefly discuss the distribution of yes/no responses for the issues after the 1st and 2nd scoring and in the final 20 issues.

8. Add a discussion as to whether the addition of devil's advocates to the Delphi process was helpful. Was there a correlation between Novelty and the issues proposed by the devil's advocates? What percentage of the devil's advocates' proposals made it through each round of cutoffs, and what percentage to the final list?

9. Report how different the ranking of the 41 issues selected in the 1st round was compared to the ranking of the issues after the 2nd round. This can signal controversy in the choices and expose whether any 'black swans' survived the scrutiny.

---

## [Author Response]

[We repeat the reviewers’ points here in italic, and include our replies in plain text].

Essential revisions:1. Restructure the Introduction so that it starts with a discussion of why the scan is needed, why Bioengineering in particular is of interest, and what gap the work is trying to fill. Limit the discussion of the methods to the end of the Introduction.

We now open the Introduction with a paragraph on why bioengineering is of interest (due to the speed of change and depth and breadth of societal impacts) and why the scan is needed (to create a process of periodically updated and improving horizon-scans as is done in other areas such as global conservation).

As recommended, we have shifted the discussion of methods to the end of the Introduction.

2. Table 1 and the review of the 20 issues are results, so a Results section should be included with these elements.

This has been added directly after the Introduction.

3. Give a more detailed (but still brief) explanation about how the scoring of the issues works in the Introduction, as well as a more detailed explanation in the Materials and Methods. Also include why 20 was chosen as a cutoff for the number of issues finally reported. Please provide (either in the article or as supplementary data) a list of the 83 issues initially proposed and the Z-scores calculated.

We have added detail to the Introduction to further explain the voting protocol.

We have tried to be as thorough as possible in providing further details on scoring in the Methods section (under ‘Phase 2: Scoring’). We now provide further information on the novelty scores, the logistics of scoring, and how long the participants were given. If there is anything further that the reviewers would like to see covered in scoring, then we would be happy to add it in.

We now explain why 20 was chosen as a cut-off in the ‘Phase Three’ section: “The decision to keep the list to 20 was made by consensus by the workshop group and was influenced by a significant difference between the z-scores of the top and bottom 20 issues, but a much smaller spread of scores within the top 20.”

We are deep believers in transparency and thus all of the anonymised scoring data (including z-scores for the long-list, short-list and final list) have already been included via the Open Science Framework (see https://osf.io/ej8p2/). We will also provide these to the editors to include as supplementary data.

4. For each of the sections describing one of the issues (currently in the Introduction, to be moved to a Results section, see above), clearly identify in each what the emerging issue is, why it's a (potential) issue (positive or negative), what the drivers of the issue are, and the current state of the issue to provide a stronger takeaway message about why each issue is important.

We have reviewed all the issues and made edits to ensure that these meet each of the recommended dimensions.

5. On the comparison between the 2017 and 2020 scan:i) Expand on the differences between the 2017 and the 2020 scan, including discussing whether any of the emerging issues of 2017 have come into effect; or why the experts may have switched focus (including whether expert selection may have biased this change; or whether any disruptive influences or drivers may have changed experts' focus).ii) Map the final issues from the 2017 scan horizon to the full list of 83 issues of the 2020 scan horizon using the Z score matrix, rather than sharing the 2017 issues as a table. If the table is not openly accessible in the previous publication, include it here as supplementary data.iii) If possible, designate which new issues on the 2020 list are due to a wider scope definition of bioengineering, otherwise the comparison between the 2017 and the 2020 scan horizons can be biased.

i) Unfortunately, providing a comprehensive analysis of what issues have come into effect requires a separate methodology and journal article. However, we have tried to highlight some specific issues which appear to have become more prominent since 2017.

ii) We have added the z-scores for the 2017 list to the supplementary materials as suggested. As noted above, the z-scores and other scoring data have already been provided via the Open Science Framework and will be sent to the editors as supplementary data as well. We would suggest that since the z-scores are being provided in the supplementary data, that the summary table of the 2017 scan be kept. This is intended as an easily digested summary which will allow for the audience to compare issues across the two scans. The discussion of similarities and differences across the scans will be much less fruitful without this qualitative summary.

iii) This is a very good idea. We have flagged now which issues we believe would have fallen outside of the 2017 scan’s scope but made it into the 2020 scan: neuronal probes expanding new sensory capabilities and the governance of cognitive enhancement.

6. Report the gender/geographical/disciplinary bias in the 2nd as compared to the 1st experts' panel, since the 1st panel had 38 participants while the 2nd panel had 26.

We have added in Table 3 which provides a comparative analysis of both groups in terms of gender balance, disciplinary distribution and geographical coverage and sample size. We make a brief mention of this in-text (under Method: Phase III) to highlight that the two groups were largely comparable across each of the characteristics.

7. Share statistics regarding the 'Novelty' yes/no response, and briefly discuss the distribution of yes/no responses for the issues after the 1st and 2nd scoring and in the final 20 issues.

We now provide statistics on the novelty of all 41 shortlist issues in Table 4 at the end of the Methods section (now Supplementary file 2). The final paragraph of the Methods section now discusses the distribution of novelty.

8. Add a discussion as to whether the addition of devil's advocates to the Delphi process was helpful. Was there a correlation between Novelty and the issues proposed by the devil's advocates? What percentage of the devil's advocates' proposals made it through each round of cutoffs, and what percentage to the final list?

We have added a paragraph to the end of “Phase II: Scoring”, directly after our initial discussion of the devil’s advocates. As recommended, this covers the statistics on the novelty and success of the issues proposed y devil’s advocates.

9. Report how different the ranking of the 41 issues selected in the 1st round was compared to the ranking of the issues after the 2nd round. This can signal controversy in the choices and expose whether any 'black swans' survived the scrutiny.

A comparison of the ranks after first and second round scoring is now provided in Table 4 (now Supplementary file 2) at the end of the Methods section. We further discuss these in the final paragraph of this section.